# Optimal Sr-Doped Free TiO₂@SrTiO₃ Heterostructured Nanowire Arrays for High-Efficiency Self-Powered Photoelectrochemical UV Photodetector Applications

**Shiming Ni, Fengyun Guo \*, Dongbo Wang \* , Shujie Jiao, Jinzhong Wang, Yong Zhang, Bao Wang and Liancheng Zhao**

Department of Optoelectronic Information Science, School of Materials Science and Engineering, Harbin Institute of Technology, Harbin 150001, China; nishiming1989@163.com (S.N.); shujiejiao@hit.edu.cn (S.J.); jinzhong_wang@hit.edu.cn (J.W.); yongzhang@hit.edu.cn (Y.Z.); baowang@stu.hit.edu.cn (B.W.); lczhao@hit.edu.cn (L.Z.)

\* Correspondence: guowen@hit.edu.cn (F.G.); wangdongbo@hit.edu.cn (D.W.); Tel.: +86-1384-509-7528 (F.G.); +86-1310-166-0586 (D.W.)

**Abstract:** Due to their high performance, photoelectrochemical ultraviolet (UV) photodetectors have attracted much attention, but the recombination of photogenerated electrons at the interface of photoanode/electrolyte limited further improvement of photoelectrochemical UV photodetectors (PEC UVPDs). Modification of TiO₂ photoanode by SrTiO₃ could improve the performance of UVPD, because the energy barrier that is established at the TiO₂–SrTiO₃ interface could accelerate the separation of the photogenerated electrons-holes pair. However, the recombination center that is caused by the preparation of TiO₂@SrTiO₃ core-shell heterostructured nanostructure decreases the performance of PEC UVPDs, which is still an important problem that hindered its application in PEC UVPDs. In this paper, we presented a Sr-doped free TiO₂@SrTiO₃ core-shell heterostructured nanowire arrays as a photoanode for the self-powered PEC UVPD. This will not only accelerate the separation of the photogenerated electrons-holes pair, but it will also reduce the recombination of photogenerated electron-hole pairs in the photoanode. The intrinsic effect of SrTiO₃ reaction time on the *J* variations of UVPDs is investigated in detail. An impressive responsivity of 0.358 A W$^{-1}$ was achieved at 360 nm for the UVPD based on TiO₂@SrTiO₃ core-shell heterostructured nanowire arrays, which heretofore is a considerably high photoresponsivity for self-powered photoelectrochemical UVPDs. Additionally, this UVPD also exhibits a high on/off ratio, fast response time, excellent visible-blind characteristic, and linear optical signal response.

**Keywords:** Sr–doped free; TiO₂@SrTiO₃; self-powered; photoelectrochemical; UV photodetector

---

## 1. Introduction

Due to important application in environmental monitoring and surveillance, missile warning, chemical and physiology analyses, fire detection, and secure communication, the precise detection of ultraviolet (UV) radiation is desperately in demand [1]. Semiconductor UV photodetectors (UVPDs) can be categorized into two types: photoconductive and photovoltaic. The photoconductive UVPDs cannot operate without extra power sources, which is applied to urge the photogenerated carriers to generate a photocurrent [2–6]. This largely enlarges the device size and weight, which leads to the disadvantages of miniaturization, wireless applications, and intellectualization.

Recently, the demand for UV photodetectors that could work without additional power sources has become more and more urgent [7–11]. A series of self-powered UVPDs have been reported and they draw a great deal of attention on account of their numerous merits [12–17]. This type of UVPD has been designed by utilizing the photovoltaic effect. According to the charge separation features of the interface, the UVPDs that are based on the photovoltaic effect can be divided into three main types: p–n junction, Schottky junction, and photoelectrochemical (PEC) cell. For the PEC-type UVPDs, a simple physicochemical process, which could reduce the preparation process complexity when compared with p-prepares n junction and Schottky junction UVPDs, prepare photoanodes and counter electrodes. Furthermore, PEC UVPDs could generate a larger current, up to mA, which could be easily detected [18–22]. Based on the preparation of photoanode on flexible substrates [23], the PEC UVPDs could have a wider application range due to their weight reduction and better mechanical flexibility [24]. In summary, research on PEC UVPDs has great significance.

Researchers have used different materials, micro-/nano-structures, electrolytes, counter electrodes, and even varied device structures to assemble PEC UVPDs to enhance the performance of UV detecting. To date, most efforts to improve the performances of PEC UVPDs has concentrated on developing photoanodes. Diverse nanostructures were employed for fabricating photoanode in PEC UVPDs, including nanoparticle (NP), nanosheet (NS), nanorod (NR), nanotube (NT), nanowire (NW), and nano forest (NF) [25]. The electron transport rates and the specific surface area (SSA) of these nanostructures mentioned above cannot be gotten simultaneously. Conventional nanocrystals photoanode require thermal sintering to strengthen the electronic contact between each particle, thus leading to added processing overhead, hindering the application of flexible substrates [26]. By contrast, one-dimensional (1D) nanostructure semiconductors, such as NWs, are candidates with great promise for high performance PEC UVPDs, because a 1D nanostructure could provide direct an electron transport tunnel to accelerate the separation of carriers to reduce the possibility of the recombination of photogenerated electrons [27–30]. Xie and coworkers reported the self-powered UVPDs based on 1D $TiO_2$ nanorod arrays. The photosensitivity of UVPDs are 0.22 $A \cdot W^{-1}$, which are greatly enhanced when compared with the nanocrystalline $TiO_2$ film based UVPD [17]. Thanks to the great developments in fabrication techniques, most semiconductor NWs could have been successfully synthesized [31,32].

Meantime, the energy gap of the photoanode materials for PEC UVPDs should be larger than 3.1 eV, relating to the wavelength of ultraviolet radiation. In the past several decades, the investigated wide bandgap oxide and sulfide nanomaterials, including ZnO [20], ZnS [33], $TiO_2$ [34], $SnO_2$ [35], and perovskite structure oxides, like $SrTiO_3$ [36], can be possibly employed as photoanode materials for PEC UVPDs. Among the various wide bandgap semiconductors, $TiO_2$ is an n-type direct bandgap semiconductor with a proper bandgap and it has been widely investigated, owing to its UV absorption features [2,37].

A good deal of 1D $TiO_2$ nanostructures have been applied as the photoanodes for PEC UVPDs [38–42]. The research of Cao et al. [40] suggests that a proper size of NRAs is essential for a higher photocurrent. Chen's group employed vertical rutile $TiO_2$ NRAs on PEC UVPDs, showing the responsivity of 25 $mA \cdot W^{-1}$ at 350 nm with a fast photoresponse [38]. Subsequently, they prepared branched $TiO_2$ nanostructures that were based on $TiO_2$ NRAs by a facile two-step chemical synthesis process. The PEC UVPDs based on branched $TiO_2$ nanostructures showed the $R_\lambda$ of 186.5 $mA \cdot W^{-1}$ at 365 nm with a low response time (the rise time of 0.15 s and the delay time of 0.05 s) Though, the superior light scattering characteristics of the branched $TiO_2$ NRAs issue in higher harvesting of the optical light region is detrimental to the spectral selectivity of UVPD.

Commonly, the approaches for improving the performance of PEC UVPDs comprise three aspects: augmenting light capturing in UV region in photoanodes, raising charge separation or suppressing charge recombination at the interface of photoanode-electrolyte, and enhancing carrier transport [43]. Charge recombination at the interface of photoanode-electrolyte is a critical problem that exists in PEC UVPD, and the loss of photogenerated electrons leads to low responsivity [44]. Building semiconductor core–shell structures with type-II band alignment is a good way to suppress charge recombination at

the interface of the photoanode-electrolyte, which has attracted plenty of interest for application as the photoanode of next generation photoelectrochemical cells. Semiconductor core–shell structures with type-II band alignment could establish a potential barrier at the interface of core–shell structures to suppress the recombination of the photogenerated electrons and the oxide of electrolyte and accelerate the separation of carriers. [45,46]. According to the energy levels that could form an energy potential barrier at the interface of photoanode-electrolyte, for $TiO_2$ nanomaterials, the core–shell structures of $TiO_2$@MgO [47], $TiO_2$@InO [48], $TiO_2$@$WO_3$ [49], and $TiO_2$@$SrTiO_3$ show prodigious probability in high-efficiency PEC cells.

The performance of $TiO_2$@$SrTiO_3$ core-shell heterostructured nanostructures could be ulteriorly improved by employing the piezo-phototronic effect. As a general matter, elastic strain was introduced during the growth of epitaxial dissimilar materials core–shell nanostructures, such as nanowire arrays (NWAs) [50,51]. As a result, a static and internally built strain that is introduced by the lattice-mismatch can create piezoelectric polarization in the nanostructure [52,53]. The piezoelectric polarization in the core–shell NWAs is beneficial to the separation of the photogenerated electron-hole pairs.

The hydrothermal method is the most economical and practical for large size devices among the various effective methods for preparing $TiO_2$@$SrTiO_3$ core-shell heterostructured NWAs. In general, $TiO_2$ NWAs and $Sr^{2+}$ source was used to prepare the $TiO_2$@$SrTiO_3$ core-shell heterostructured NWAs. The reaction from $TiO_2$ to $SrTiO_3$ could be regarded as a simultaneous decomposition–deposition procedure that includes the decomposition of $TiO_2$ followed by the deposition of the $SrTiO_3$. When the break of $TiO_2$ lattice occurred at the surface of $TiO_2$ NWAs, the $Sr^{2+}$ could diffuse into the lattice of $TiO_2$ to form the Sr-doped $TiO_2$@$SrTiO_3$ heterostructures arrays. The doped $Sr^{2+}$ may bring more recombination centers and oxygen vacancies in the $TiO_2$@$SrTiO_3$ heterostructure and damage the carrier transport pathway that is provided by the $TiO_2$ nanowires. These lead to more recombination of photogenerated electron and the degradation of the of the PEC UVPD performance [54].

To solve the doped $Sr^{2+}$ in $TiO_2$@$SrTiO_3$ heterostructures, in this paper, we present the $Sr^{2+}$-doped free $TiO_2$@$SrTiO_3$ core-shell heterostructured NWAs as a photoanode for the self-powered PEC UVPD. When compared with the UVPD that is based on the bare $TiO_2$ NWAs, the UVPD based on $Sr^{2+}$-doped free $TiO_2$@$SrTiO_3$ core-shell heterostructured NWAs exhibits a higher responsivity and sensitivity (on/off ratio) under UV light irradiance. The effect of various preparation conditions of $TiO_2$@$SrTiO_3$ core-shell heterostructured NWAs is discussed in detail. Subsequently, the spectral-response characteristics, photo-sensitivity, response speed, and light intensity dependence of the *J* signal are also tested to estimate the performance of UVPD based on $Sr^{2+}$–doped free $TiO_2$@$SrTiO_3$ core-shell heterostructured NWAs.

## 2. Results and Discussion

The experimental section is shown in Supplementary Data. The morphology of pure $TiO_2$ and $TiO_2$@$SrTiO_3$ heterostructured nanowire arrays (STO0, STO1, STO2, STO3, and STO4) was characterized by field emission scanning electron microscope (FESEM), as shown in Figure 1. The top view of pure $TiO_2$ NWAs, as shown in Figure 1a, illustrates that the cross section of nanowires is square with an average side length of 15 nm due to the tetragonal structure of the rutile $TiO_2$, and the nanowire arrays exhibited regularly distribution. The cross-sectional FESEM image (Figure 1b) of STO0 shows that the nanowire arrays are perpendicular to the substrate with an approximate 33 μm thickness. Figure 1c–j indicate that the growth of $SrTiO_3$ on the surface of $TiO_2$ NWAs does not alter the diameter and thickness of $TiO_2$ NWAs, which indicates that the growth of $SrTiO_3$ happens in-situ on the surface of the $TiO_2$ nanowire. To investigate the elemental distribution of $TiO_2$@$SrTiO_3$ heterostructures nanowire arrays, energy-dispersive X-ray spectroscopy (EDS) mapping characterization is employed (STO3 sample). Supplementary materials Figures S1 and S2 show the EDS elemental mapping images of the cross section and top surface of $TiO_2$@$SrTiO_3$ heterostructured nanowire arrays (STO3), respectively. Due to the homogeneous distribution of Sr element among the cross section and top surface of $TiO_2$@$SrTiO_3$ heterostructures nanowires, we can ensure that

the formation of $SrTiO_3$ happens on the surface of all $TiO_2$ nanowires, so as to form $TiO_2@SrTiO_3$ heterostructure nanowires arrays.

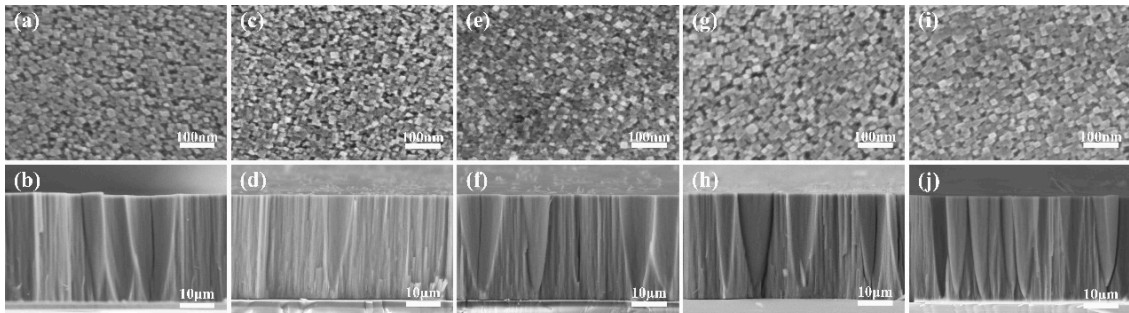

**Figure 1.** Morphological characterizations of the smooth-surface and rough-surface TiO2 NWAs: (**a,c,e,g,i**) top-surface FESEM image of STO0, STO1, STO2, STO3, STO4, respectively; and (**b,d,f,h,j**) cross-sectional FESEM image of STO0, STO1, STO2, STO3, STO4, respectively.

To figure out the existential of the $SrTiO_3$ coating layer, the detailed information regarding the microscopic structure of $TiO_2@SrTiO_3$ heterostructured nanowires (STO2) are exposed by transmission electron microscopy (TEM), as shown in Figure 2. The structure of $TiO_2@SrTiO_3$ heterostructured nanowire cluster is shown in the low magnification TEM image (Figure 2a), which indicated that the uniform nanowires, with a diameter of about 15 nm, constitute the nanowire cluster, as is consistent with the SEM results. The porous structure that each nanowire separates from the other makes for the immersion of the electrolyte. Figure 2b displays the high-resolution TEM (HRTEM) lattice image of a single $TiO_2@SrTiO_3$ heterostructured nanowire. The distinct lattice fringes differentiate $TiO_2$ and $SrTiO_3$ from the high-resolution TEM image. The obvious lattice spacing of 0.325 nm is in accordance with that of the (002) plane of rutile $TiO_2$, indicating that the growth direction of rutile $TiO_2$ NWs is along [001]. The interplanar spacing, with 0.272 nm, corresponds well to that of the (110) plane of cubic SrTiO3, which demonstrates the formation of a $TiO_2@SrTiO_3$ heterojunction structure. The $SrTiO_3$coating layer is uniformly distributed and about 1.5 nm.

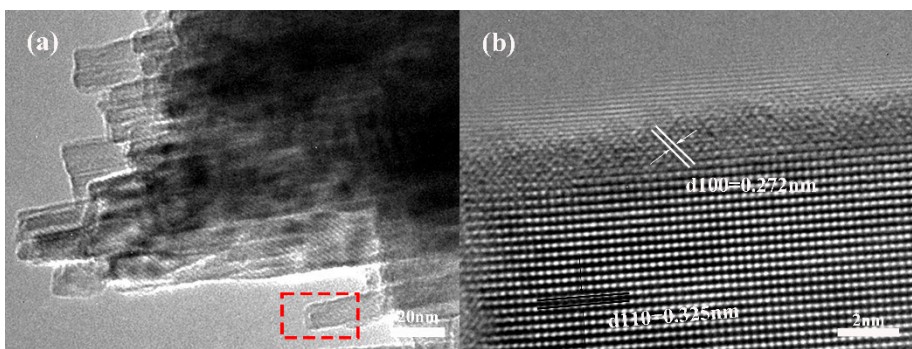

**Figure 2.** (**a**) TEM image of nanowire bundles; (**b**) HRTEM image of a single $TiO_2@SrTiO_3$ heterostructured nanowire.

The X-ray diffraction (XRD) result of the as-prepared $TiO_2$ nanowire arrays through hydrothermal treatment in PH 14 $Sr(OH)_2$ aqueous solution with different treatment time (STO1, STO2, STO3, and STO4 samples). The crystalline structures and the state of Sr-doped of pure $TiO_2$ and $TiO_2@SrTiO_3$ heterostructured nanowire arrays were investigated by X-ray diffraction (XRD) in the 2θ range of 30–80°. The XRD patterns of pure $TiO_2$ (STO0) and $TiO_2@SrTiO_3$ heterostructured nanowire arrays (STO1, STO2, STO3, STO4) are shown in Figure 3a. For pure $TiO_2$ nanowire arrays (STO0), the obvious diffraction peak that is located at 62.72° originates from the (002) plane of the tetragonal rutile phase $TiO_2$ (JCPDS 21-1276), which should be attributed to the verticality of $TiO_2$ nanowire arrays of highly

[001] oriented growth and Fluorine-doped tin oxide (FTO) substrates. The XRD patterns did not show a distinctive peak of $SrTiO_3$ phase, which might be due to the minor content of $SrTiO_3$ as compared to $TiO_2$, which can be effortlessly understood that the Bragg reflections that were indexed to $SrTiO_3$ were not detected. There is something different that could be found in the magnification of XRD pattern of the (002) peak that was prepared at different treatment time, as shown in Figure 3b. It should be emphasized that, when the reaction time is short, the consistent location of (002) peaks of STO0, STO1, and STO2 suggest that no Sr element was doped into $TiO_2$ lattice. When the treatment time is longer, the gradual shifting of the (002) peak occurred, which could be observed in the STO3 and STO4 samples. The shift of (002) peak location to a small angle suggests that the interplanar distance of (002) lattice plane augments due to the increase of treatment time. As the size of $Sr^{2+}$ is larger than that of $Ti^{4+}$, the augment of interplanar distance can be attributed to the replacement of $Ti^{4+}$ by $Sr^{2+}$. The state of Sr element doping into the lattice of $TiO_2$ nanowire array can be further examined by the following X-ray photoelectron spectroscopy (XPS) techniques.

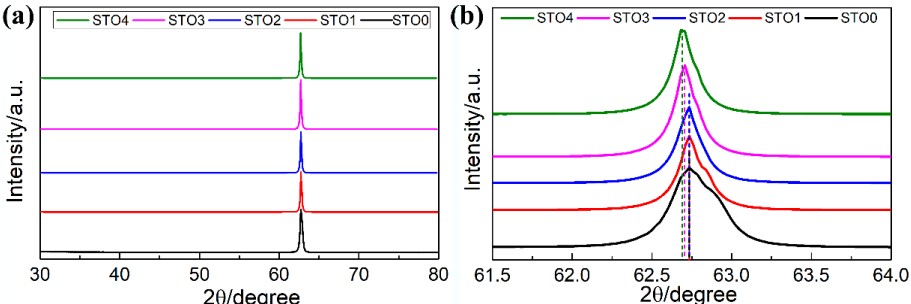

**Figure 3.** (**a**) X-ray diffraction (XRD) patterns of pure $TiO_2$ (STO0) and $TiO_2@SrTiO_3$ heterostructured nanowire arrays, (**b**) the enlarged X-ray diffraction (XRD) patterns between $61.5°$ and $64.0°$.

More specifics about the surface chemical state of $TiO_2@SrTiO_3$ heterostructured nanowire are explored by using XPS. From fully scanned spectrum (Figure S3a), the existence of Sr, Ti, O, and C in the $TiO_2@SrTiO_3$ heterostructured nanowire is confirmed. The peak of C 1s could be attributed to the carbon based impurity, and the binding energy for C 1s at 284.6 eV is applied as a criterion to calibrate the binding energies of the other elements [55]. The spectrum of the Ti 2p (Figure S3b) shows that the characteristic peak positions center at 458.28 eV related to Ti 2p3/2 and 463.98 eV related to Ti 2p1/2. The peak that is centered in 458.28 eV is comprised of $TiO_2$ Ti 2p3/2 (458.6 eV) and $SrTiO_3$ Ti 2p3/2 (457.8 eV), and the peak that is centered in 458.28 eV is comprised of $TiO_2$ Ti 2p1/2 (464.4 eV) and $SrTiO_3$ Ti 2p1/2 (463.5 eV) [56,57]. This could approve the formation of $TiO_2@SrTiO_3$ heterostructure. The XPS spectral peak of O 1s shows two small asymmetry peaks, (Figure S3c) one centered at 529.36 eV that should be linked with $TiO_2$ and $SrTiO_3$, meanwhile, the other peak that is centered at 531.08 eV is believed to be correlated with oxygen vacancy [58,59]. At the same time, a typical XPS fitted profile of Sr 3d is shown in Figure 4. The high resolution XPS spectrum of the Sr 3d region can be decomposed into two peaks of Sr 3d5/2 and Sr 3d3/2, which are centered at 132.74 and 134.47 eV, respectively. The peak at 132.74 eV is in accordance with those that were reported for perovskite $SrTiO_3$, and the energy peak at 134.47 eV can be attributed to SrO complexes, which is attributed to the Sr element doping in the lattice of rutile $TiO_2$ [60]. As shown in Figure 4a,b, when the reaction time is short, there is no peak in the high resolution XPS spectrum of the Sr 3d related to SrO, which means that no Sr element was doped into $TiO_2$ nanowires. With the reaction time increasing, the energy peak that is related to SrO appeared, which means the Sr was doped into $TiO_2$ nanowires, as shown in Figure 4c,d.

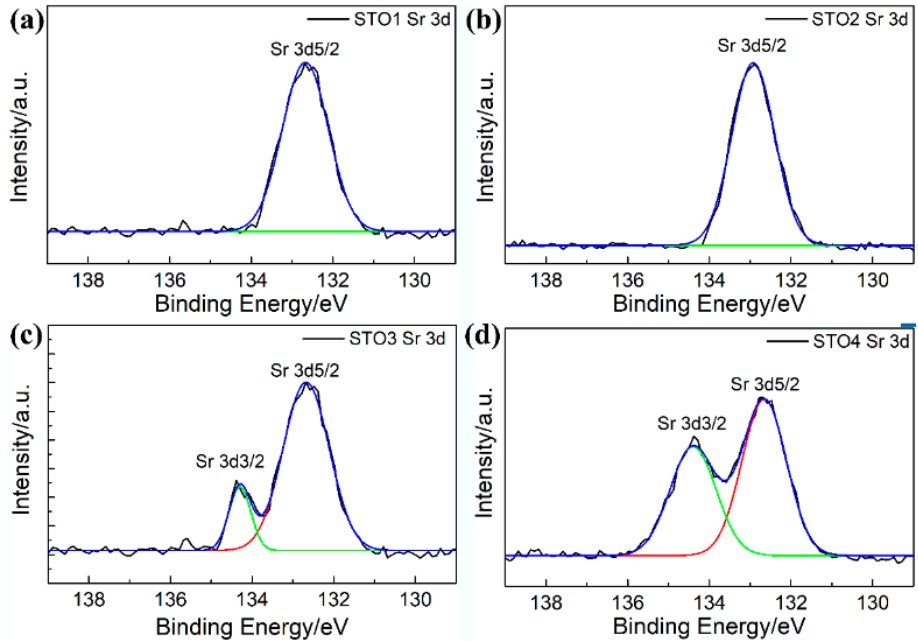

**Figure 4.** X-ray photoelectron spectroscopy (XPS) spectra of Sr 3d of TiO2@SrTiO3 heterostructured nanowire arrays: (**a**) STO1, (**b**) STO2, (**c**) STO3, (**d**) STO4.

To evaluate the performance of the photoelectrochemical UVPDs based on STO0, STO1, STO2, and STO3 electrodes, the photocurrent-density ($J$) via voltage ($V$) characteristics were measured under a 10 mW·cm$^{-2}$ UV light source whose wavelength is 365 nm. A comparison of *J-V* characteristics of all UVPDs is shown in Figure 5a, and the *J-V* characteristics of UVPD based on STO4 is shown in Figure S4. The detailed data about the short-circuit photocurrent density ($J_{sc}$), the open-circuit photovoltage ($V_{oc}$), fill factor (FF), and power conversion efficiency ($\eta$) are listed in Table 1. With the reaction time ranging from 0 to 60 min (STO0 to STO3), the $J_{sc}$ first increases from 1.50 mA·cm$^{-2}$ (STO0), to a maximum value of 3.48 mA·cm$^{-2}$ (STO2), and then decreases with further increasing the treatment time. The $V_{oc}$ keeps an increasing trend with the increase of treatment time. The UVPDs that are based on STO0 show the lowest $V_{oc}$ due to the intrinsic property of TiO$_2$ pure nanowires. The FF values that were obtained from the UVPDs show little change that suggests the growth of SrTiO$_3$ do not destroy the well contact between pure TiO$_2$ nanowires film and the FTO glass substrate. The power conversion efficiency ($\eta$) is the maximum of the product of photo current density and photo voltage, and FF could be calculated by the following equation: FF = $\eta / (J_{sc} \times V_{oc})$. The $\eta$ displays the same trend with $J_{sc}$, which is enough to drive low-power-consuming complementary metal-oxide semiconductor (CMOS) circuits and certain nanodevices [61,62].

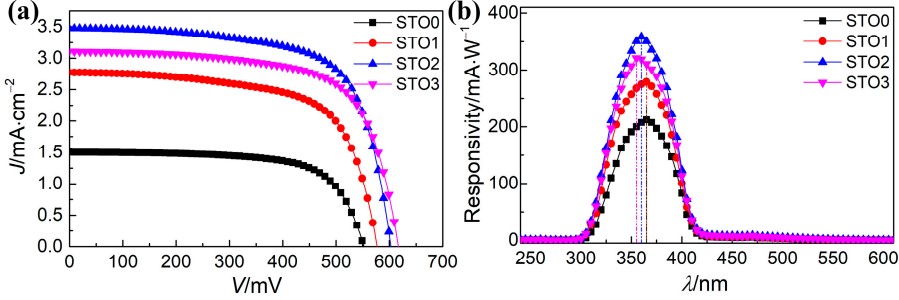

**Figure 5.** (**a**) *J–V* characteristics and (**b**) the spectral photoresponsivity of the photoelectrochemical ultraviolet photodetectors (PEC UVPDs) based on STO0, STO1, STO2, and STO3.

**Table 1.** Detailed data of short-circuit photocurrent density ($J_{sc}$), open-circuit photovoltage ($V_{oc}$), fill factor (FF), power conversion efficiency ($\eta$), and photoresponsivity of all UVPDs.

| Sample | $J_{sc}$ (mA·cm$^{-2}$) | $V_{oc}$ (mV) | FF (%) | $\eta$ (%) | Photoresponsivity (mA·W$^{-1}$) |
|--------|------|------|------|------|------|
| STO0 | 1.50 | 552 | 68.1 | 0.566 | 212 at 365 nm |
| STO1 | 2.77 | 577 | 67.9 | 1.045 | 279 at 365 nm |
| STO2 | 3.48 | 602 | 67.7 | 1.418 | 358 at 360 nm |
| STO3 | 3.11 | 616 | 67.6 | 1.294 | 321 at 355 nm |

To pinpoint the cause of the $J_{sc}$ variation, we tested the spectral photoresponsivity of the photoelectrochemical UVPDs based on STO0, STO1, STO2, and STO3. The relationship between the responsivity (defined as photocurrent per unit of incident optical power) and the incident light wavelength is shown in Figure 5b. As presented in Figure 5b, the incident light wavelength ranges from 250 to 600 nm at 0 bias. For the photoelectrochemical UVPD that is based on STO0, at 365 nm, the peak value of photoresponsivity is about 212 mA·W$^{-1}$, corresponding to the forbidden bandwidth of pure rutile TiO$_2$. The FTO could absorbed the illumination whose wavelength is shorter than 300 nm, at the same time, photoresponsivity sharply reduces by about three orders of magnitude when the illumination whose wavelength is longer than 420 nm, which is proper for visual-blind ultraviolet detection application. For the photoelectrochemical UVPD that is based on TiO$_2$@SrTiO$_3$ heterostructured nanowire arrays (STO1, STO2, and STO3), the peak values of photoresponsivity are 279 at 365 nm, 358 at 360 nm, and 321 at 355 nm, respectively. The blue shift of the peak position may be due to that the forbidden bandwidth of SrTiO3 is larger than that of pure rutile TiO$_2$. The variation tendency of photoresponsivity is consistent with the $J_{sc}$.

In fact, two factors that include the light harvesting efficiency and the rate of electron recombination mostly influence the photoresponsivity of the PEC UVPDs [63].

UV-vis diffuse reflectance spectra measurement was carried out to confirm the effect of the reaction time on the light harvesting efficiency. The thickness of nanowire arrays is too large, which meant that no light could pass through the nanowire arrays. The less reflection indicates more absorption. As displayed in Figure S5a, the variation the reflection of the pure TiO$_2$ nanowire arrays (STO0) and TiO$_2$@SrTiO$_3$ heterostructured nanowire arrays (STO1, STO2, STO3, STO4) is regular. With the reaction time increasing, the absorption of samples enhances at a range of wavelengths from 300 to 400 nm. The equation for calculating the optical forbidden bandwidth from the diffuse reflectance is shown in Supplementary Information [64]. As shown in Figure S5b, the optical forbidden bandwidth of samples (STO0, STO1, STO2, STO3, STO4) with different reaction times is estimated to be 2.99, 3.02, 3.06, 3.09, and 2.95 eV, respectively. For the samples STO0, STO1, STO2, and STO3, the result of the optical forbidden bandwidth is consistent with the result of photoresponsivity, but for the sample with long reaction time (STO4), the doped Sr element into the crystal lattice of rutile TiO$_2$ could create an impurity energy level and narrow the band gap [54], which accounts for the self-doping effect due to the bonding character of Ti–O–Sr [65]. Accordingly, the optical forbidden bandwidth is the result of the comprehensive effect of the SrTiO$_3$ coating and doped Sr element. The distinct variation of the optical forbidden bandwidth also indicates that the amount of doped Sr element of STO4 obtained is larger than that of the sample STO3.

Electrochemical impedance spectroscopy (EIS) is applied at a bias of 0.5 V under dark condition to depict the photogenerated electron–hole pairs recombination dynamics at the photoanode/electrolyte interface to investigate the effect of the reaction time on the rate of electron recombination. Figure 6a displays the Nyquist plot curves of photoelectrochemical UVPDs that are based on STO0, STO1, STO2, and STO3. On the basis of the EIS model, in the Nyquist plot curve, the diameter of the obvious semicircle in the middle-frequency region, represents the charge transfer at the photoanode/electrolyte interface [66,67]. Accordingly, the diameter of the obvious semicircle increases with the reaction time augment, indicating that the possibility of the recombination of the photogenerated electrons and the redox in electrolyte decrease with the reaction time augment. This result is consistent with the

variation tendency of photoresponsivity of photoelectrochemical UVPDs that are based on sample based on STO0, STO1, and STO2, but not with that of photoelectrochemical UVPDs that are based on sample based on STO3. Figure 6b shows plots of the interfacial charge recombination resistance as a function of the applied bias voltage, indicating the same results.

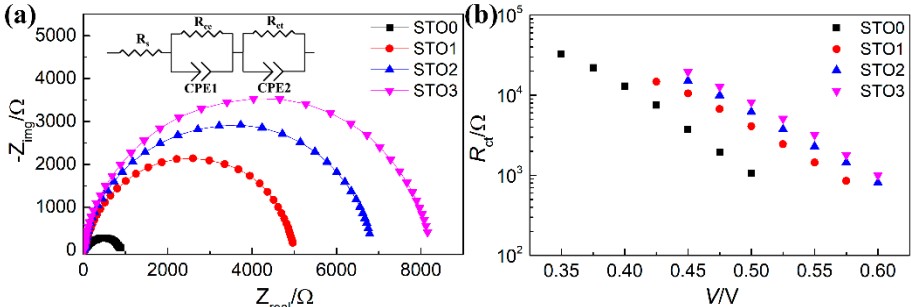

**Figure 6.** (**a**) Electrochemical impedance spectra of UVPDs based on STO0, STO1, STO2, and STO3 measured at 0.5 V bias voltage in dark condition (**b**) Plots of the inter- facial charge recombination resistance as a function of the applied bias voltage.

The dark current test was carried out to simply understand the photogenerated electrons recombination condition. Although the dark current is not direct data to provide the electron recombination rate, but lower dark current suggests lower photogenerated electrons recombination. As shown in Figure S6, at a same potential, the dark current of photoelectrochemical UVPD based on sample STO2 is the smallest among all of the UVPDs, which indicates that the rate of the recombination of photogenerated electrons is the smallest.

To identify the effect of reaction time on the photogenerated lifetime to understand the condition of photogenerated electron recombination open-circuit voltage decay (OCVD) more deeply, measurements of the UVPDs that were assembled with different photoanode were carried out. The OCVD technique has been commonly applied to examine the exciton electron lifetime ($\tau_e$) and photogenerated electrons recombination problem in UVPDs [68–70]. The $\tau_e$ can be calculated from the corresponding OCVD curve of the UVPDs (as shown in Figure S7) and the calculation formula is also shown in Supplementary Information. Seeing that there is no current moving through the external circuit at open circle condition, the electrons must be reacted by $I_3^-$ ions at the photoanode-electrolyte interface. Thus, the slow decay of $V_{oc}$ must be owed to the slow photogenerated electron recombination with the oxidation state of electrolyte on at the photoanode-electrolyte interface. The calculated $\tau_e$ as a function of $V_{oc}$ (in log-linear representation) is shown in Figure 7a. According to the previous research result [69–72], the $\tau_e$ value at a high voltage region mainly reflects the electron lifetime in the photoanodes. As illustrated in Figure 7a, the $\tau_e$ variation trend resembles the variation tendencies of photoresponsivity of photoelectrochemical UVPDs, which indicates that the photogenerated electrons are not only recombined with the redox in the electrolyte.

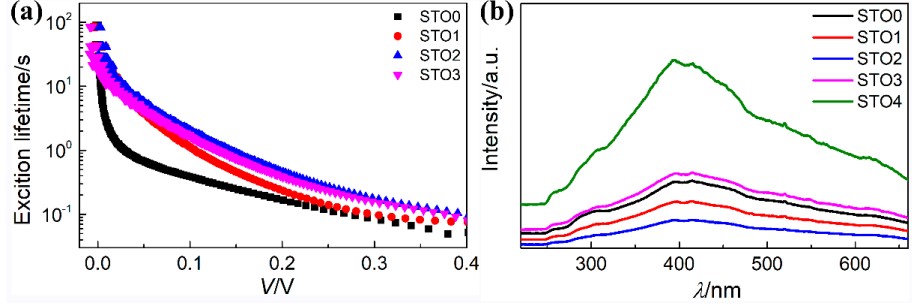

**Figure 7.** (**a**) the $\tau_e$ (in log-linear representation) as a function of open-circuit voltage, (**b**) the photoluminescence (PL) spectra measured at room temperature of all samples on FTO substrates.

Due to the photogenerated carrier recombination being capable of leading to a noteworthy emission signal in the photoluminescence (PL) spectrum, the photogenerated charge carriers' separation behavior of the photogenerated electron-hole pairs can be appraised by measurements of PL emission [73]. The PL intensity variation indicates the change of the recombination property of photogenerated electron-hole pairs in the photoanode. Figure 8 shows the PL spectra of pure $TiO_2$ nanowire arrays (STO0) and $TiO_2$@$SrTiO_3$ heterostructured nanowire arrays (STO1, STO2, STO3, STO4). We could be informed that the reaction does not qualitatively change the shape of the PL emission, nevertheless, the PL intensity of the $TiO_2$@$SrTiO_3$ heterostructured nanowire arrays was found to be meaningfully varied when compared to that of the pure $TiO_2$ nanowire arrays (STO0). After the $TiO_2$@$SrTiO_3$ heterostructure is formed, the intensity of PL peaks significantly decreases. The established energy barrier caused by the difference of the energy level of core-shell structures with type-II band alignment can help in the separation of the photogenerated electron-hole pairs and thus suppress interfacial charge recombination that decreases the probability of photogenerated carrier's recombination in the photogenerated to produce photoluminescence. The obvious decrement of PL intensity can be attributed to the reduced recombination rate of photogenerated electron-hole pairs. With the further increasing of reaction time, the intensity of the PL peak regularly enhances. Doped $Sr^{2+}$ elements could bring a defect of crystal lattice and recombination centers in the $TiO_2$@$SrTiO_3$ heterostructured nanowire arrays. At the same time, doped $Sr^{2+}$ elements damage the carrier transport pathway that is built by the $TiO_2$ nanowire. Both of these would bring out more recombination of photogenerated electron–hole pairs and then enhance the intensity of PL emission [54].

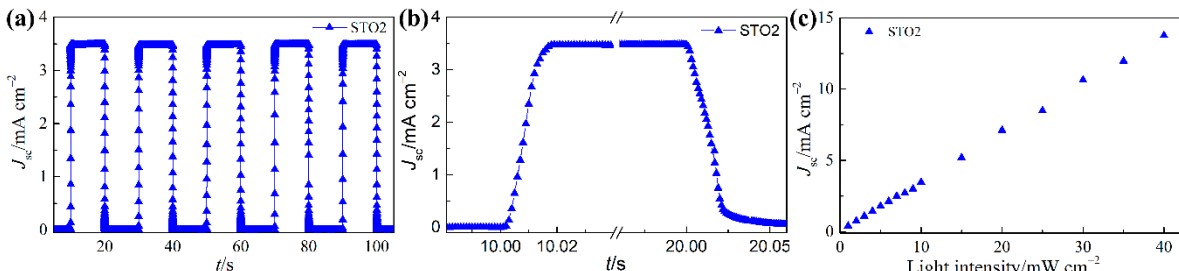

**Figure 8.** (**a**) Photocurrent responses, (**b**) The enlarged rising and decaying edges of the photocurrent response, and (**c**) J as a function of the incident UV light intensity from 1 to 40 mW·cm$^{-2}$ of UVPD based on STO2.

In common, evaluating the performance of UVPD from several perspectives, such as photoresponsivity, repeatability, stability, response speed, correlation between photocurrent and light intensity, and so on. To estimate the repeatability and stability of all UVPDs, time-dependent photocurrent signal is investigated under periodic pulse UV radiance with a luminous power density of 10 mW·cm$^{-2}$, 20 s switching period, and 365 nm wavelength at zero bias, as shown in Figure 8a and Figure S8a. The cyclically regular switch of photocurrent signal from the "on" state to the "off" state indicates the ample repeatability and stability of this type of UVPDs. The sensitivity, also called on/off ratio, of the J signal can reach 28064 for the UVPD based on STO2, which is considerably higher than that (6276) of the UVPD that is based on STO0.

In common, the response speed was described by the rise time $\tau_r$ (the time to reach (1-1/e) of the maximum J) and decay time $\tau_d$ (the time for the photocurrent to drop to 1/e of its maximum J). Enlarged rising and decaying edges of the photocurrent response of all UVPDs are shown in Figure 8b and Figure S8b. the detailed data of $\tau_r$ and $\tau_d$ are listed in Table 2. We can find that the $\tau_r$ is very close to each other (11ms), but the $\tau_d$ of UVPD that is based on STO2 is the highest. In fact, the speed of recombination of photogenerated electron determines the magnitude of $\tau_d$. In this experiment, the biggest factor that affects the speed of recombination of the photogenerated electron is that the photogenerated electron combines with the $I_3^-$ in the electrolyte at the interface of the photoanode/electrolyte. From previous research in this manuscript, we can know that

the recombination of photogenerated electron with the $I_3^-$ in the electrolyte at the interface of photoanode/electrolyte is suppressed. Therefore, UVPD that is based on the STO2 sample achieves the longest decay time, although it presents the largest on/off ratio and photoresponsivity, which is consistent with the tendency of $\tau_e$. To investigate the reaction between $J_{sc}$ and the intensity of incident light, the $J_{sc}$ as a function of the intensity of incident light of all UVPDs is presented in Figure 8c and Figure S8c. The $J_{sc}$ measurements of UVPDs were measured under a 365 nm UV light source, whose light intensity varied from 1 to 40 mW·cm$^{-2}$. Obviously, the $J_{sc}$ shows an outstanding linear relationship with the growing intensity of incident light in a wide range. The slope of data is fitted through linear approximation by the least-square method, as shown in Figure S9. Moreover, the slope of $J_{sc}$ and the intensity of incident light matches the $R_\lambda$ that is attained from Figure 5c., suggesting that this type of photodetector is suitable in the precise measurement of intensity of UV irradiation.

**Table 2.** Dark current, on/off ratio, τr and τd of UVPDs based on STO0, STO1, STO2, STO3.

| Sample | Dark Current (mA·cm$^{-2}$) | On/Off Ratio | $\tau_r$ (ms) | $\tau_d$ (ms) |
|--------|------------------------------|--------------|---------------|---------------|
| STO0 | $2.39 \times 10^{-4}$ | 6276 | 10 | 12 |
| STO1 | $1.53 \times 10^{-4}$ | 18104 | 11 | 13 |
| STO2 | $1.24 \times 10^{-4}$ | 28064 | 11 | 15 |
| STO3 | $1.36 \times 10^{-4}$ | 22867 | 11 | 14 |

## 3. Conclusions

In summary, Sr-doped free TiO$_2$@SrTiO$_3$ core-shell heterostructured nanowire arrays were successfully synthesized on FTO glass. We explored the effect of reaction time on the preparation of TiO$_2$@SrTiO$_3$ core-shell heterostructured nanowire arrays. When compared with pure TiO$_2$ nanowire arrays, the photocurrent of UVPDs have been significantly enhanced after coating with SrTiO$_3$. The UVPDs with a reaction time 40 min exhibited more excellent photoresponse properties than other TiO$_2$@SrTiO$_3$ heterostructured nanowire arrays that are based UVPDs. The electrochemical impedance spectroscopy results show that the formation of TiO$_2$@SrTiO$_3$ heterostructures can efficiently reduce the recombination at the interface of photoanode/electrolyte, and the photoluminescence results show the Sr doped enhance the recombination of photogenerated electron-hole pairs in the photoanode. Two effects dominate the performance of the UVPDs together. A large responsivity of 358 mA·W$^{-1}$ was achieved at 360 nm for the UVPD based on Sr-doped free TiO$_2$@SrTiO$_3$ heterostructured nanowire arrays (reaction time is 40 min) without applied bias, together with the excellent on/off ratio of 28064. Additionally, this UVPD also exhibits fast response speed, excellent visible-blind characteristic, and linear optical signal response. Due to its operability in actual application, all of these manifest that this type of UV photodetector is a considerable selective candidate as next-generation UVPD.

**Supplementary Materials:** The following are available online at http://www.mdpi.com/2073-4352/9/3/134/s1, Experimental method, Figure S1: EDS elemental mapping images of cross section of TiO$_2$@SrTiO$_3$ heterostructured nanowire arrays (STO3), Figure S2: EDS elemental mapping images of top surface of TiO$_2$@SrTiO$_3$ heterostructured nanowire arrays (STO3), Figure S3: (a) XPS of TiO$_2$@SrTiO$_3$ heterostructure nanowire arrays (STO2), (b) XPS spectra of Ti 2p, and (c) XPS spectra of O 1s, Figure S4: *J–V* characteristics a of the PEC UVPDs based on STO4, Figure S5: (a) The UV-vis diffusive reflectance spectra of samples STO0, STO1, STO2, STO3, STO4 and (b) the plots of transforming the Kubelka–Munk function versus the energy of light, Figure S6: Dark current density–voltage of all UVPDs (a) normal, (b) semi-logarithmic plots, Figure S7: Open-circuit voltage decay for all UVPDs and the UV intensity at first 10 s is 10 mA·cm$^{-2}$, Figure S8: (a) Photocurrent responses, (b) The enlarged rising and decaying edges of the photocurrent response, and (c) *J* as a function of the incident UV light intensity from 1 to 40 mW·cm$^{-2}$ of UVPDs based on STO0, STO1, STO2, STO3, Figure S9: Linear fitting of *J* as a function of the incident UV light intensity from 1 to 40 mW·cm$^{-2}$ of UVPDs based on STO0, STO1, STO2, STO3.

**Author Contributions:** F.G. and D.W. designed the experiments; S.N. and B.W. performed the experiments; S.N., Y.Z., J.W. and S.J. analyzed the data; Y.Z., J.W. and S.J. contributed analysis tools; S.N. and D.W. wrote the paper, L.Z. coordinated the overall work.

**Funding:** This research was funded by the National Natural Science Foundation of China (Grant No. 51502061) and (Grant No. 61605036).

**Conflicts of Interest:** The authors declare no conflict of interest.

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
