# Peer review of "Optimal Sr-Doped Free TiO2@SrTiO3 Heterostructured Nanowire Arrays for High-Efficiency Self-Powered Photoelectrochemical UV Photodetector Applications"

_crystals, doi:10.3390/cryst9030134_

Round 1
Reviewer 1 Report
In Figure 7(b), the NW STO4 shows the highest PL. But the J-V is not shown in Figure 5. Can the author add it in Figure 5?
It will be better for the author to the synthesis conditions and the structural parameter of the NW heterostructures (STO0, STO1, STO2, STO3, and STO4) to Table 1 and Table 2 for comparison.
Author Response
Response to Reviewer 1 Comments
Point 1: In Figure 7(b), the NW STO4 shows the highest PL. But the J-V is not shown in Figure 5. Can the author add it in Figure 5?
It will be better for the author to the synthesis conditions and the structural parameter of the NW heterostructures (STO0, STO1, STO2, STO3, and STO4) to Table 1 and Table 2 for comparison.
Response 1: Thank you for your careful work. We would like to add the J-V curve of the PEC UVPD based on STO4 in the Supplementary Materials. In fact, the Jsc of the PEC UVPD based on STO3 is lower than that of STO2, so the tendency of the Jsc with the amount of Sr doping is obvious.

Reviewer 2 Report
First of all، I see this article original and suitable for publication in "Crystals". However, the English language is poor and the whole article need to be professionally re-written as there are plenty of errors. here are some examples and comments I ask the authors to reply to them one by one before final acceptance:
Line 2: The title: should be changed to be: "Optimal Sr-free TiO2@SrTiO3 ............'
Line 16: word "attion" should be corrected to "attention"
Line 22: There is a space between "the" and "performance"
Lie 28: put "on" after "based"
Line 37:"applications" not "application"
page2
Line 41: "which is to ..." should be corrected to "which leads to ....."
line 46: delete "on" from "....by utilizing on the photovoltaic effect ..."
Line 59: ...(ND) should be corrected to (NR)
line 63: " ... leading added processing ..." should be corrected to "leading to added processing ...."
Line 63 through line 66: be more specific why one dimensional nanostructures have higher performance than nanoparticles and give examples for the high performance applications.
Line 89: Why Core -shell structures with type II band alignment are preferred in photoelectrochemical applications ? give more explanations.
Line 103 through line 113: paragraph need to be grammatically revised and scientifically re-written.
line 114:" Sr-doped free TiO2@SrTiO3 ..." should be changed to "Sr-free Tio2@SrTiO3 ..." This phrase should be corrected through the whole manuscript.
line 136: the word "clarify" should be changed to "investigate"
Line 144: "... the existential state of ..." should be changed to " .. the existence of ..."
Line 185 the sentence beginning wth " The spectrum of the Ti2p .....through "..... Heterostructure" in line 187 is incomplete and need to be rewritten.
Line 205: Authors should mention how they measured or calculated the power conversion efficiency and mention the formula used to calculate it.
More and detailed information about the experimental procedures and conditions must be added to the experimental methods section in supplementary data file
Author Response
Response to Reviewer 2 Comments
First of all, I see this article original and suitable for publication in "Crystals". However, the English language is poor and the whole article need to be professionally re-written as there are plenty of errors. here are some examples and comments I ask the authors to reply to them one by one before final acceptance:
Point 1: Line 2: The title: should be changed to be: "Optimal Sr-free TiO2@SrTiO3 ............'
Response 1: Thank you for your valuable advice. We have changed the title as your advice.
Point 2: Line 16: word "attion" should be corrected to "attention"
Response 2: Thank you for your careful work. We are sorry for the spelling mistake, and have corrected in the manuscript.
Point 3: Line 22: There is a space between "the" and "performance"
Response 3: Thank you for your careful work. We are sorry for the spelling mistake, and have corrected in the manuscript.
Point 4: Line 37:"applications" not "application" page2
Response 4: Thank you for your careful work. We have corrected in the manuscript.
Point 5: Line 41: "which is to ..." should be corrected to "which leads to ....."
Response 5: Thank you for your careful work. We have corrected in the manuscript.
Point 6: line 46: delete "on" from "....by utilizing on the photovoltaic effect ..."
Response 6: Thank you for your careful work. We have corrected in the manuscript.
Point 7: Line 59: ...(ND) should be corrected to (NR)
Response 7: Thank you for your careful work. We have corrected in the manuscript.
Point 8: line 63: " ... leading added processing ..." should be corrected to "leading to added processing ...."
Response 8: Thank you for your careful work. We have corrected in the manuscript.
Point 9: Line 63 through line 66: be more specific why one dimensional nanostructures have higher performance than nanoparticles and give examples for the high performance applications.
Response 9: Thank you for your valuable advice. We have explained why one dimensional nanostructures have higher performance than nanoparticles as follow :“By contrast, 1D nanostructure semiconductors such as NWs are candidates with great promise for high performance PEC UVPDs, because 1D nanostructure could provide direct electron transport tunnel to accelerate the separation of carriers to reduce the possibility of the recombination of photogenerated electrons [27-30].” We have added the examples in the manuscript as follow: “Xie and coworkers reported the self-powered UVPDs based on 1D TiO2 nanorod arrays. The photosensitivity of UVPDs are 0.22 A W–1, which are greatly enhanced compared with the nanocrystalline TiO2 film based UVPD”. And the modification in the manuscript has been marked by blue.
Point 10: Line 89: Why Core -shell structures with type II band alignment are preferred in photoelectrochemical applications? give more explanations.
Response 10: Thank you for your advice. We have added more explanations about “why core-shell structures with type II band alignment are preferred in photoelectrochemical applications” as follow: “Building semiconductor core–shell structures with type-II band alignment is a good way to suppress charge recombination at the interface of photoanode-electrolyte, which has attracted plenty of interest for the application as the photoanode of next generation photoelectrochemical cells. Semiconductor core–shell structures with type-II band alignment could establish a potential barrier at the interface of core–shell structures to suppress recombination the photogenerated electrons and the oxide of electrolyte and accelerate the separation of carriers. [45,46].” And the modification in the manuscript has been marked by blue.
Point 11: Line 103 through line 113: paragraph need to be grammatically revised and scientifically re-written.
Response 11: Thank you for your advice. We have rewritten as follow: “The hydrothermal method is the most economical and practical for large size devices among various effective methods for preparing TiO2@SrTiO3 core-shell heterostructured NWAs. In general, TiO2 NWAs and Sr2+ source was used to prepare TiO2@SrTiO3 core-shell heterostructured NWAs. The reaction from TiO2 to SrTiO3 could be regarded as a simultaneous decomposition–deposition procedure that includes the decomposition of TiO2 following by deposition of the SrTiO3. When the break of TiO2 lattice occurred at the surface of TiO2 NWAs, the Sr2+ could diffuse into the lattice of TiO2 to form the Sr-doped TiO2@SrTiO3 heterostructures arrays. The doped Sr2+ may bring more recombination centers and oxygen vacancies in the TiO2@SrTiO3 heterostructure and damage the carrier transport pathway providing by the TiO2 nanowires. These lead to more recombination of photogenerated electron and degradation of the of the PEC UVPD performance.” And the modification in the manuscript has been marked by blue.
Point 12: line 114:" Sr-doped free TiO2@SrTiO3 ..." should be changed to "Sr-free Tio2@SrTiO3 ..." This phrase should be corrected through the whole manuscript.
Response 12: Thank you for your advice. In this manuscript, we solved the problem that the doping Sr in TiO2 NWAs would trap the photogenerated electrons and decrease the Jsc. But it is not that no Sr exist in TiO2@SrTiO3 heterostructured NWAs. So we could not change “Sr-doped free TiO2@SrTiO3 ...” to “Sr-free Tio2@SrTiO3”.
Point 13: line 136: the word "clarify" should be changed to "investigate"
Response 13: Thank you for your advice. We have changed in the manuscript.
Point 14: Line 144: "... the existential state of ..." should be changed to " .. the existence of ..."
Response 14: Thank you for your advice. We have changed in the manuscript.
Point 15: Line 185 the sentence beginning with " The spectrum of the Ti2p .....through "..... Heterostructure" in line 187 is incomplete and need to be rewritten.
Response 15: Thank you for your advice. We have changed in the manuscript as following sentence: “The spectrum of the Ti 2p (Figure S3b) shows that the characteristic peak positions center at 458.28 eV related to Ti 2p3/2 and 463.98 eV related to Ti 2p1/2. The peak centered in 458.28 eV is comprised of TiO2 Ti 2p3/2 (458.6 eV) and SrTiO3 Ti 2p3/2 (457.8 eV), and the peak centered in 458.28 eV is comprised of TiO2 Ti 2p1/2 (464.4 eV) and SrTiO3 Ti 2p1/2 (463.5 eV). This could approve the formation of TiO2@SrTiO3 heterostructure.” And the modification in the manuscript has been marked by blue.
Point 16: Line 205: Authors should mention how they measured or calculated the power conversion efficiency and mention the formula used to calculate it. More and detailed information about the experimental procedures and conditions must be added to the experimental methods section in supplementary data file
Response 16: Thank you for your valuable advice. We have added the method to calculate the power conversion efficiency. More and detailed information about the experimental procedures and conditions have been added to the experimental methods section in supplementary data file. And the modification in the manuscript has been marked by blue.
